# Crack Detection and Analysis of Concrete Structures Based on Neural Network and Clustering

**DOI:** 10.3390/s24061725

**Published:** 2024-03-07

**Authors:** Young Choi, Hee Won Park, Yirong Mi, Sujeen Song

**Affiliations:** 1Earth Turbine, 36, Dongdeok-ro 40-gil, Jung-gu, Daegu 41905, Republic of Korea; youngch5@naver.com; 2School of Architecture, Civil, Environment and Energy Engineering, Kyungpook National University, 80, Daehak-ro, Buk-gu, Daegu 41566, Republic of Korea; phoid1@knu.ac.kr

**Keywords:** deep learning, concrete crack detection, Resnet50, k-means clustering

## Abstract

Concrete is extensively used in the construction of infrastructure such as houses and bridges. However, the appearance of cracks in concrete structures over time can diminish their sealing and load-bearing capability, potentially leading to structural failures and disasters. The timely detection of cracks allows for repairs without the need to replace the entire structure, resulting in cost savings. Currently, manual inspection remains the predominant method for identifying concrete cracks. However, in today’s increasingly complex construction environments, subjective errors may arise due to human vision and perception. The purpose of this work is to investigate and design an autonomous convolutional neural network-based concrete detection system that can identify cracks automatically and use that information to calculate the crack proportion. The experiment’s findings show that the trained model can classify concrete cracks with an accuracy of 99.9%. Moreover, the clustering technique applied to crack images enables the clear identification of the percentage of cracks, which facilitates the development of concrete damage level detection over time.

## 1. Introduction

Structural Health Monitoring (SHM) is dedicated to continuous monitoring and assessing the structural integrity of various infrastructure systems, with its primary objective being the identification and evaluation of any changes or damages that may occur over time. Despite the typical design lifespan of civil engineering structures being several decades (ranging from 50 to 100 years), with concrete often constituting around fifty percent of building structures, various factors contribute to the formation of cracks in concrete construction, including temperature fluctuations, concrete shrinkage, overloading, long-term fatigue stresses, and cyclic loading. These cracks have significant implications, compromising the stability and robustness of concrete elements, therefore decreasing the structure’s ability to support loads and creating hazards to general stability and safety. The rapid detection and repair of cracks in buildings are essential to preserve their structural integrity and safety [1].

Crack detection on construction sites is still predominantly conducted manually, employing two primary methods: destructive and non-destructive testing. Both approaches utilize measurement instruments and visual inspection to evaluate surface conditions for defects. However, depending solely on visual inspection can be labor-intensive and time-consuming. The subjective nature of quantitative analysis means that results heavily rely on the expertise of specialists. Furthermore, the labor-intensive process of visual inspection often leads to delayed fracture assessments, especially in hard-to-reach areas, potentially causing setbacks in crucial maintenance tasks. To enhance the precision and efficiency of inspections, there is a growing demand for advanced technological tools and automated systems [2]. To swiftly and reliably analyze surface defects, researchers have developed automated image-based crack detection. Compared with manual procedures, this approach uses image processing techniques to generate reliable results. Various image formats are used in this context, including RGB, infrared, ultrasonic, laser, and time-of-flight diffraction [3]. Automating the image-based crack detection process using computer vision techniques in image processing has proven effective. Incorporating extracted manual characteristics into neural networks, support vector machines, random forests, and other conventional machine learning techniques enhances their accuracy. However, manual feature-based fracture identification techniques remain susceptible to noise and changes in lighting [4]. To address the aforementioned challenges, researchers are increasingly turning their attention to the field of deep learning. Studies on deep learning-based crack detection fall into three main categories: object identification, semantic segmentation, and image classification [5]. Although numerous studies have used various types of crack datasets, relatively few have employed thermal imaging cameras for crack image collection. In addition, most existing studies have focused on concrete structures with uniform backgrounds, which limits their applicability in complex façade scenes. Therefore, effective strategies are imperative to minimize the detrimental impacts of background noise on crack detection and enhance the accuracy and robustness of the detection process.

The purpose of this study is to introduce an automated approach that utilizes hybrid images, combining RGB and thermal data, and employs a convolutional neural network (CNN) for concrete crack detection. This is intended to enhance existing research methodologies. Initially, this study focuses on improving image quality through image processing techniques while implementing region of interest (ROI) technology to eliminate interference from image noise. Subsequently, a deep learning model is applied for the classification of damage in concrete structures. Following that, k-means clustering is utilized to identify the proportion of cracks, enabling inspectors to monitor trends in crack damage at any given time. This innovative method holds promise in improving the efficiency of concrete structure inspections by providing real-time insights into crack evolution for timely intervention and maintenance.

The remaining content in the document is arranged as follows: Section 2 introduces related studies; Section 3 discusses and describes the proposed methodology; Section 4 presents the model, assessment outcomes, and interpretations; and Section 5 summarizes this study.

## 2. Related Work

### 2.1. Concrete Crack Detection

Figure 1 categorizes crack detection methods into two primary groups: learning-based approaches and image processing-based techniques. This classification establishes a coherent organizational framework for comprehending the diverse methodologies employed in the field of crack detection. By delineating these overarching categories, the image offers a thorough summary of all the different methods used in the field of fracture detection.

Image processing-based crack detection primarily involves preprocessing, edge detection, image percolation, and morphological approaches [6]. A fracture detection technique based on the Otsu algorithm was proposed in [7]. This approach employs multiple filters, applies suitable thresholds to distinguish between the background and foreground, and identifies significant cracks using the Otsu method. This method demonstrates the ability to accurately and distinctly detect cracks in images. In [8], a percolation modeling-based improved crack identification technique was proposed. This refinement effectively eliminates noise regions, thereby enhancing the detection accuracy. For safety monitoring, ref. [9] introduced an automated technique for identifying and categorizing cracks in subway tunnels. High-speed CMOS cameras captured tunnel surface images, and cracks were isolated from the grayscale background using morphological and thresholding procedures. Eliminating unnecessary noise in the image enhanced the final detection rate and accuracy. Another study, ref. [10], introduced a method for the rapid and accurate identification of cracks in concrete walls by analyzing images captured with a digital camera. To enhance the images and improve the results, a median filtering technique was employed. The proposed image processing method with metaheuristic optimization provides a feasible substitute for automatically identifying concrete cracks.

In [11], a method using thermography was proposed to detect cracks in welds, along with determining the geometrical characteristics and orientation of the cracks to aid in predicting the direction of crack expansion. However, the technique lacked sufficient accuracy for measurements in the transverse direction compared with conventional methods. Recently, crack detection has been shifting toward the use of machine learning algorithms to enhance the efficacy of image-based crack inspection techniques. Similarly, ref. [12] presented a novel technique that uses multiscale neighborhood data and pixel intensity to automatically detect pavement cracks at the pixel level. The proposed technique uses probabilistic generative modeling (PGM) and pixel intensity data to calculate the likelihood of a fracture emerging at each pixel. Initially, a probability map is generated by calculating the probability that each pixel represents a portion of a fracture using methods based on probabilistic generative models (PGMs). Subsequently, a weighted expansion operation is introduced to enhance the identification of boundary pixels, thereby improving crack continuity and resulting in higher crack detection performance. In [13], a crack detection algorithm that leverages a trained stochastic classifier to distinguish between cracked and uncracked images was proposed. Compared with the Otsu binarization method, the proposed method exhibited faster processing speed than model-based detectors such as the percolation model, facilitating rapid and accurate coarse extraction. However, the presence of noise in concrete structures poses challenges. Extracting effective features using the crack feature model in complex environments can be challenging, resulting in a less reliable detection model. In [14], an automatic fracture detection technique based on artificial neural networks was proposed for concrete construction. The process begins with the classification of patches containing cracks in the input image, followed by crack segmentation. Subsequently, crack features such as length and width are analyzed using image processing techniques. Another study [15] employed convolutional neural networks to develop a detector for automatic crack identification from images of concrete constructions. However, the detection accuracy was reduced by the influence of concrete surface stains.

In [16], a deep CNN (DCNN) trained on ImageNet data was used to automatically identify fractures in Portland cement concrete (PCC) and hot-mix asphalt (HMA). However, it encountered challenges in distinguishing between concrete pavement cracks and joints. Notably, this study did not use data augmentation, which affected the classifier’s accuracy. In [17], a method for automatically classifying image blocks cropped from 3D road images using deep CNNs was proposed. All four proposed CNNs achieved an overall classification accuracy exceeding 94% after effective training on four distinct sensory field-size supervised CNNs. Nevertheless, further studies are needed to investigate the effects of other hyperparameters on the CNNs. Another study, ref. [18], introduced Potspot, a deep learning-based method for detecting potholes. Postpot offers real-time crack identification with end-to-end mapping. In terms of performance, the model achieved a higher accuracy (approximately 97.6%) compared with three pre-trained models (InceptionV3, VGG-19, and VGG-16) and six baseline approaches (ANN, SVM, and K-nearest neighbors (KNN)). A deep full convolutional network (FCN)-based method for concrete crack detection, leveraging semantic segmentation of crack images, was proposed in [19]. The proposed approach demonstrated effective crack detection and accurate crack density estimation, achieving an average accuracy of approximately 90% for the FCN network.

Another study, ref. [20], investigated a deep learning method for road crack detection, comparing faster R-CNN and Mask R-CNN. Although both methods accomplished crack detection with just over 130 images, Mask R-CNN exhibited reduced accuracy in bounding box detection. In [21], deep learning was introduced to harness UAVs for bridge crack detection. Leveraging UAVs and faster R-CNN offers potential for improved precision and efficiency in automated bridge crack detection. Similarly, ref. [22] proposed integrating you only look once (YOLO) with unmanned aerial vehicles (UAVs) for real-time crack detection in tiled sidewalks. Several network topologies were evaluated, including YOLOv2-tiny, YOLOv3-tiny, ResNet50-based YOLOv2, and Darknet19-based YOLOv2. RestNet 50-based YOLOv2 and YOLOv4-tiny achieved outstanding accuracies of 94.54% and 91.74%, respectively, while demonstrating an exceptional ability to rapidly detect even small cracks. Another study, ref. [23], proposed a model for structural defect detection leveraging AlexNet, VGGNet13, and ResNet18 models. Tests revealed strong performance from ResNet18, while the YOLOv3 model achieved superior accuracy in crack region detection. A common limitation of deep learning-based models is their reliance on extensive and annotated datasets, making them susceptible to performance degradation in the presence of noise or background interference. In addition, the significant computational time required by these models is a notable concern.

### 2.2. Crack Clustering

Leveraging clustering techniques in crack identification offers substantial automation opportunities for crucial tasks in crack data analysis. By uncovering underlying structures and patterns within data, extracting essential features, accurately pinpointing crack positions, recognizing diverse types of cracks, streamlining data preprocessing, and enabling data-driven decision support for inspections, cluttering techniques play a crucial role in automating various aspects of crack data analysis in construction projects. In [24], CrackLG, an autonomous system for detecting dam cracks that leverages local-global clustering analysis was proposed. Crack region delineation was achieved through a multi-step process involving preprocessing, local clustering, binarization, and global clustering analysis. Dam surface cracks can be accurately and swiftly detected using image analysis, eliminating the need for manual intervention. In [25], an enhanced rail crack signal detection method is proposed using the acoustic emission (AE) technique, lever-aging joint optimized clustering and time window features. This method employs a joint optimization clustering approach using an LSTM codec network and k-means clustering. Experimental results demonstrate the enhanced approach’s efficacy in accurately identifying rail crack signals despite significant noise interference. In addition, ref. [26] proposed a crack detection approach that leverages convolutional neural networks and fuzzy spectral clustering. The proposed method achieved superior detection accuracy compared with existing methods. Integrating the system onto a robot enabled it to access and assess cracks directly, achieving in situ measurements. In [27], an unsupervised approach to crack classification by combining convolutional neural networks with k-means clustering was proposed. The technique achieved satisfactory performance with average accuracy values of 0.806, 0.792, and 0.913 for transverse, longitudinal, and alligator cracks, respectively. Another study, ref. [28], suggested using density-based spatial clustering of applications with noise (DBSCAN), an unsupervised clustering technique, in conjunction with acoustic emission monitoring to monitor crack damage in mild steel. In [29], an unsupervised learning approach for fracture detection using wavelet clustering of accelerometer data collected from smartphones was proposed. Although low-cost smartphone sensor data may not capture the finer details of pavement fractures, they offer a viable solution for detecting road pavement breaks and distinguishing between different types of pavement faults. In another study [30], a four-scale detection layer YOLOv3-FDL model was proposed to pinpoint minute cracks in ground-penetrating radar images. The model K-means++ clustering to determine suitable anchor boxes, leading to improvements in the mean average precision (mAP) and training speed. However, a notable limitation is the relatively small dataset of the collected radar images. Current techniques for detecting cracks based on clustering techniques grapple with several challenges such as feature selection, data noise, complex-shaped cracks, missing labels, algorithm selection, large-scale data, and real-time processing.

## 3. Materials and Methods

This study proposes an automated technique for concrete crack detection using a convolutional neural network. This network efficiently eliminates unnecessary background noise, followed by a clustering technique applied to the detected crack images to output the percentage of cracks. The model leverages a hybrid concrete crack dataset that comprises both RGB and thermal images, thereby enhancing feature richness, adaptability, robustness, and extensiveness in crack detection.

The proposed research program, illustrated in Figure 2, comprises the following steps: (a) the collection and organization of data into datasets through experiments, followed by data preprocessing and filtering; (b) the enhancement of image quality and the elimination of image background interference through image processing techniques; (c) employing deep learning models for detection and classification; (d) the identification of crack regions through clustering techniques and outputting crack occupancy; and (e) the analysis of crack detection using confusion matrices and other evaluation indices, considering different deep learning model classification performances and the clustering effect after image processing. Each step is elaborated upon in detail below.

### 3.1. Data Collection

The datasets employed in the proposed model were collected from the concrete building complex of Kyungpook University and the construction site of Kyungil University in Korea. Summer is more conducive to thermal imaging image acquisition than winter. This is primarily attributed to the elevated temperatures during summer, which result in in-creased temperature differentials between objects. This heightened thermal contrast allows thermal imaging to more clearly capture the infrared radiation emitted by objects, thereby enhancing the accuracy of detecting and analyzing thermal features [31]. Furthermore, the warmer climate aids in improving the visibility of infrared signals, facilitating the easier capture and rendering of these signals using thermal imaging equipment. Therefore, we chose to collect the dataset during the summer. Figure 3 illustrates the data collection site. Detailed technical specifications for the high-resolution thermal camera used in the experiment are outlined in Table 1. In addition, Table 2 provides information on the placement height of the thermal camera, the distance from the thermal camera to the target, and the horizontal angle of the thermal camera during the data observation in the experiment.

In this study, a close-up data gathering strategy was employed to categorize concrete photos into two groups: those with cracks and those devoid of fractures. After filtering out ineligible images, a dataset of 1500 thermal images was compiled for analysis. To ensure consistency and streamlined processing, the image dimensions were standardized to 100 × 100 pixels. This standardization involved compressing the image resolution and enhancing the quality using a dedicated image conversion software. The uniformity achieved in image size facilitates consistent processing and enables the implementation of the proposed automated concrete crack detection technique. Figure 4 illustrates a reference sample image from the dataset.

### 3.2. Algorithm

The proposed work is systematically detailed in the following sections, each addressing a specific phase of the research. The initial step involves image preprocessing, which enhances the quality of the original images. Subsequently, various image processing methods are applied to individually improve each image. The final phase involves assessing the classification accuracy of the test images. A CNN automatically classifies the images. Four processing modules were used for the implementation in this study.

Step 1: Image preprocessing. To ensure consistent dimensionality before feeding into the neural network, raw images in the dataset were initially resized to 100 × 100 pixels using MATLAB R2020b’s “imresize” function. Subsequently, two image preprocessing techniques (Flip and Rotation) were applied to significantly augment the number of images in the dataset. This data augmentation mitigates overfitting issues that may arise during neural network training, thereby enhancing the model’s generalization capabilities for new data. The processing results are depicted in Figure 5, yielding a dataset of 4500 images.

Step 2: Grayscale. Using MATLAB’s “rgb2gray” function, the resulting color images were converted to grayscale. This processing step eliminates the hue and saturation information from the image while preserving the brightness information. In the present study, converting from a color image to a grayscale enhances crack visibility by altering the pixel value comparison depth. The processing step is illustrated in Figure 6.

Step 3: Edge detection techniques. Edge detection is a pivotal process in image analysis that efficiently reduces data volume while preserving essential information within the image. This technique focuses on identifying the prominent characteristics of edges in an image, allowing for the extraction of key features and patterns. Edge detection highlights intensity or color changes between image regions, enabling a focused representation of the critical elements within the visual data. This stage extracts key edge features from the image using various edge detection techniques to generate a distinct edge map. Using the “Sobel” technique, as seen in Figure 7, minimizes without affecting the sample image’s borders.

Step 4: Region of interest (ROI). ROI, an acronym for “region of interest,” is important when it comes to computer vision and image processing. It denotes a specific area or region within an image that is intentionally selected for further analysis or processing. Identifying and isolating an ROI instead of processing the entire image, which can be computationally expensive, enables a more targeted and efficient approach. Focusing on the region deemed most relevant enhances computational efficiency, thereby reducing processing time. In the context of this study, the cracks in the image are treated as the ROI. This approach minimizes the impact of background interference on the accuracy of the subsequent image recognition results. By focusing specifically on the region containing cracks, computational analysis becomes more targeted and image processing is optimized for concrete crack detection. The outcome of this image processing, which emphasizes the selected ROI, is illustrated in Figure 8.

### 3.3. Image Classifier

CNNs, widely used in deep learning, excel in image analysis, particularly object identification, recognition, and classification. Comprising filters, pooling layers, fully connected layers, and a Softmax function, CNNs discern intricate patterns for complex visual tasks. They excel in improving image classification with increasing depth, allowing hierarchical feature learning. However, deeper CNNs may face challenges like “gradient disappearance” or “gradient explosion”. Gradient disappearance hinders convergence with extremely small gradients, and the gradient explosion problem poses a risk of numerical instability due to exponentially growing gradients during training. These issues can impede network training, particularly with extensive training times, and contribute to the “degradation problem” of deep networks. This problem involves progressive plateauing and potential decline in network performance as the depth increases, posing a challenge in maintaining or improving accuracy.

The proposed methodology aims to improve image classification accuracy using the CNN model, particularly focusing on ResNet variants such as ResNet18, ResNet34, ResNet50, ResNet101, and ResNet152. Each variant differs in its residual module and the number of stacking times. For the evaluation, we selected ResNet50 because of its robust performance. This model, which stands for Residual Network with 50 layers, exhibits resilience against performance decline, which is often attributed to vanishing gradients. In instances where the network is excessively deep, the gradient value diminishes to 0, preventing weight updates and consequently hindering the learning process. The core concept of ResNet is to introduce an identity shortcut connection that bypasses one or more layers [32]. Skip connections, which link layers that incorporate batch normalization and ReLU, provide regularization benefits, thereby avoiding layers that negatively affect performance. Therefore, training very deep neural networks is not impeded by vanishing gradients, as is the case with conventional CNN models. Figure 9 illustrates a typical ResNet50 architecture.

After the image processing and region of interest (ROI) extraction stages, a convolutional neural network (CNN) is utilized as a feature extractor, while a support vector machine (SVM) acts as the classifier for image classification. The study employs the default Stochastic Gradient Descent (SGD) as the optimization method, setting the initial learning rate at 1 × 10^−4^ and limiting the training epochs to a maximum of 100. The classification process involves a support vector machine (SVM) with a linear kernel, and confusion matrices are generated to assess the model’s performance on both the training and testing datasets. In the model training phase, a simple neural network is constructed, incorporating convolutional layers, ReLU activation, max-pooling layers, a fully connected layer, and a Softmax classification layer. The proposed model is implemented at the network layer by Architectural Structure 1. To ensure optimal efficiency during the training phase of the experimental models, the validation loss per epoch is monitored, and weight variables are adjusted as the validation loss decreases.
**Architectural Structure 1**layers = 
   Image put   [100 100 3]   2-D Convolution   (5, 20)   ReLU
   2-D Max Pooling   (2, ‘Stride’, 2)   Fully connected   2   Softmax
   Classification output   crossentripyexOptions = trainingOptions
   sgdm
   Execution environment   CPU   Max epochs   100   Validation data   {trainingSet, testingSet}   Validation frequency   5   Initial learn rate   1 × 10^−4^   Gradient threshold   1   Verbose   false   Plots   training-progress

### 3.4. Crack Clustering

After classification by the model, datasets containing cracks were further analyzed using a crack clustering technique to determine the percentage of cracked areas. Clustering, a method employed to partition data into discernable groups, aims to organize a dataset of observations into distinct clusters. Notably, among the various clustering methods, k-means clustering is particularly prominent. Introduced by MacQueen in 1967 [33], the k-means clustering algorithm operates iteratively, continuously refining the objective function, referred to as the cluster sum of squares. The optimization process involves several crucial steps. The algorithm starts by iteratively initializing the clustering centers. Subsequently, it assigns data points to the nearest clustering centers and updates these centers to partition the data samples into a predetermined number of clusters. The essence of the algorithm lies in its iterative refinement of clustering centers, striving for compact and well-separated clusters. The algorithm minimizes the sum of squares of distances between data points and their assigned clustering centers, aiming to create meaningful and distinct clusters within the dataset. In summary, the k-means clustering algorithm iteratively refines the positions of clustering centers, optimizing the arrangement of clusters to enhance their compactness and separation. This iteration continues until convergence, resulting in well-defined and distinct clusters within the data. Suppose the data samples constitute a dataset X = {x_1, x_2, ⋯, x_n} with n samples. The objective is to partition the species into k clusters, where k must meet the following conditions: 1. Each cluster must not be empty. 2. Each sample can belong to only one cluster. Here are the steps of the k-means clustering algorithm:
Number of Clusters: Define the desired number of clusters, denoted as k, into which the dataset will be partitioned.Initialize Cluster Centers: Select k data points at random from the dataset to serve as the initial cluster centers, represented by $m_{1},{\;xm}_{2},\cdots ,m_{k}$.Iterative Process: The iterative process unfolds as follows:
(a)Allocate data points to the closest cluster center: Determine the distance between each cluster center $m_{j}$ for each data point $x_{i}$, and then allocate the data point to the cluster with the closest center. The assignment is determined by the formula $c^{\left(i\right)}=arg\;\min_{j}{\left\|x^{\left(i\right)}-m_{j}\right\|}^{2}$, where $c^{\left(i\right)}$ is the index of the assigned cluster, $x^{\left(i\right)}$ is the *i*-th data point, and $m_{j}$ is the center of the *j*-th cluster.(b)Update Cluster Centers: Update the cluster centers in accordance with the mean of each cluster’s data points. The update is performed using the formula $m_{j}=\frac{1}{n_{j}}\sum_{i=1}^{n_{j}}x_{ij}$, where $m_{j}$ is the new center of the *j*-th cluster, $n_{j}$ is the number of data points in the *j*-th cluster, and $x_{ij}$ is the *j*-th data point in the *i*-th cluster.Compute Objective Function (Sum of Squares Within Clusters): Evaluate the objective function $J=\sum_{i=1}^{k}\sum_{j=1}^{n_{i}}{\left\|x_{ij}-m_{i}\right\|}^{2}$, where $n_{i}$ represents the number of data points in the *i*-th cluster, and $m_{i}$ denotes the center of the *i*-th cluster.Output Results: If the objective function converges, output the final cluster centers $m_{1},{xm}_{2},\cdots ,m_{k}$ as the result. Otherwise, return to step 2 and iterate until convergence is achieved.

Although k-means clustering has the advantage of ease of implementation, it has notable drawbacks. First, the algorithm requires the a priori specification of the number of clusters, denoted as k. However, determining an appropriate value for k can be challenging because the optimal number of classes for dividing the dataset is often unknown. This ambiguity poses a limitation, and selecting an inaccurate value for k may compromise the clustering results. Second, the initial position of the clustering centers significantly influences the effectiveness of k-means clustering. The inappropriate selection of initial positions may result in multiple iterations, increased computational requirements, and, in some cases, convergence to local optimal solutions rather than global ones. This phenomenon can affect the accuracy of the clustering results, making the initial clustering center positions a critical consideration. At this stage, clustering for RGB images of cracks primarily focuses on crack detection. The crucial factors influencing clustering results are decisions regarding the number of clusters and the initialization of cluster centers. To address the previously mentioned shortcomings, the following modifications were implemented:Number of Clusters: Determining the optimal number of clusters is pivotal to the efficacy of the k-means clustering algorithm. The silhouette coefficient approach is employed to achieve this. The algorithm iterates through various values of k (number of clusters), computing the silhouette coefficient for each. This coefficient gauges an object’s degree of similarity to its cluster compared to others. Plotting these coefficients yields a curve graph. The k value corresponding to the highest silhouette coefficient represents the optimal number of clusters. This approach circumvents the need to manually specify the number of clusters, thereby enhancing the robustness and accuracy of the clustering results.Initialize Cluster Centers: The Otsu algorithm is used to determine the initial cluster centers in the clustering algorithm. In addition, it is employed to determine the threshold used as the filtering criterion for initializing the cluster centers. Given that concrete cracks typically involve grayscale level transitions between two adjacent regions with different grayscale levels, an appropriate threshold is derived from the average values of these two regions. Leveraging the Otsu algorithm, an efficient image segmentation method, expedites the convergence of the algorithm by selecting the threshold determined by Otsu as the initial cluster center. This strategy not only improves the quality of the initial cluster centers by achieving rapid initialization, reducing iteration times, and preventing convergence to local optima, but also leverages the advantages of the Otsu algorithm in image data processing, thereby enhancing the efficiency and accuracy of the initialization process.

## 4. Results and Discussion

### 4.1. Classifier Model Performance Analysis

A thorough assessment was carried out utilizing a confusion matrix in order to determine the efficacy of the suggested model, which incorporates four key parameters: accuracy, sensitivity, precision, and F1 score. The distinct characteristics of each parameter are visually illustrated in Figure 10. The confusion matrix, which encompasses various performance metrics, serves as a robust tool for evaluating the model’s accuracy, sensitivity (true positive rate), precision (positive predictive value), and F1 score (harmonic mean of precision and sensitivity). These metrics collectively provide a nuanced assessment of the model’s ability to accurately classify instances, minimize false positives and negatives, and achieve an optimal balance between precision and recall. The dataset was partitioned according to the distribution outlined in Table 3, with 30% allocated for model testing and 70% for model training.

This study employs four parametric equations to interpret the confusion matrix and associated data. As outlined in Table 4, the definitions for TN, TP, FN, and FP are as follows:

True Negative (TN): TN indicates that images with concrete cracks are correctly categorized as “Cracks”.

False Positive (FP): FP refers to images with concrete cracks incorrectly categorized as “Non-Cracks”.

False Negative (FN): FN signifies non-crack images incorrectly classified as “Cracks”.

True Positive (TP): TP denotes crack images correctly classified as “Non-Cracks”.

Table 5 summarizes the evaluation metrics used in this study along with their computation formulas. These metrics are essential for quantifying the performance of the proposed model. The definitions and formulas facilitate a comprehensive analysis of the model’s accuracy, sensitivity, precision, and F1 score, allowing for a nuanced understanding of its effectiveness in concrete crack detection. All of these measures work together to provide a thorough evaluation of the model’s performance, addressing various aspects of its ability to correctly classify concrete crack images.

Recall serves as a metric to gauge the model’s effectiveness in identifying positive samples, thereby enhancing its ability to recognize instances of the positive class. Conversely, precision reflects the model’s proficiency in correctly detecting negative samples. A model’s accuracy reflects its ability to correctly classify both positive and negative samples, serving as an overall measure of its performance. The F1 score serves as a balance metric that incorporates both precision and recall. A higher F1 score indicates a more reliable classification model. In addition, the model exhibits improved performance when accompanied by higher accuracy and recall scores. These metrics typically range between 0.0 and 1.0, with higher values indicating superior model performance. Striking a balance between precision and recall, the F1 score provides a comprehensive assessment of the model’s ability to correctly classify instances across different classes.

Figure 11a illustrates the model’s training set confusion matrix immediately after Sobel image processing, achieving a classification accuracy of 93.7%. In Figure 11b, the model’s test set confusion matrix is presented after Sobel image processing, resulting in a classification accuracy of 90.6%. Figure 11c shows the model’s training set confusion matrix after processing with both the Sobel operator and ROI, exhibiting an improved classification accuracy of 99.9%. Figure 11d displays the model’s test set confusion matrix after processing with both the Sobel operator and ROI, with a corresponding increase in classification accuracy to 99.9%. These results indicate that incorporating ROI effectively reduces background noise, thereby enhancing the model’s classification accuracy.

Figure 12 shows a visual representation of the training accuracy and loss after 750 iterations for two scenarios: one using solely Sobel processing and the other incorporating both Sobel and ROI processing. Regarding accuracy, the training exhibits progressive improvement, demonstrating rapid growth from the initial iterations. Conversely, the combined Sobel and ROI model rapidly stabilized and achieved near-perfect accuracy (99.9%) within just 50 iterations. This accelerated convergence achieved with ROI processing implies its contribution to a more efficient learning process. The combined Sobel and ROI model’s loss curve reaches saturation after 100 iterations, indicating stable and minimized loss. In contrast, the Sobel-only model demonstrates a more gradual loss saturation effect, implying slower convergence. Table 6 complements the visual analysis by presenting precision, sensitivity, and F1 scores for both training and testing sets of both models. By incorporating both Sobel and ROI treatments, the model achieves higher accuracy and precision, reaching an impressive 99.9% accuracy. This improvement highlights the effectiveness of ROI in enhancing the learning process.

### 4.2. Crack Clustering Model Performance Analysis

In this paper, three concrete crack images were selected as the test data for clustering, as depicted in Table 7.

#### 4.2.1. The Selection of Cluster Number

Figure 13 shows the line plots of the silhouette scores for different cluster numbers. The optimal cluster number, denoted as k, can be determined by analyzing silhouette coefficients. As illustrated in Figure 13a, which corresponds to image 1, when the cluster number k is set to 2, the silhouette coefficient reaches its highest value. Therefore, 2 was identified as the optimal cluster number for image 1. Similarly, Figure 13b represents the line plot for image 2, revealing that the silhouette coefficient attains its peak when k is 2, thereby indicating that 2 is the optimal cluster number. In Figure 13c, the line plot for image 3 is presented, which demonstrates the highest silhouette coefficient when the cluster number k equals 2. This consistent silhouette score across different figures strongly indicates that 2 is the optimal cluster number.

#### 4.2.2. Selection of the Initial Cluster Center

After determining the optimal number of clusters, initial cluster centers are selected using thresholds obtained using the Otsu method. This step is taken to address the instability issue in the clustering results. In Figure 14, the optimal thresholds for image 1, image 2, and image 3 are identified as 120, 116, and 106, respectively. Table 8 presents the optimal cluster numbers and thresholds for each image.

Achieving a mean cluster center value around neighboring thresholds results in a relatively good segmentation outcome. Observations revealed remarkably complete and well-defined clustered crack morphology. This meticulous process ensures stable and precise clustering, which contributes to reliable segmentation outcomes for each concrete crack image.

#### 4.2.3. Silhouette Coefficient

Rousseeuw [34] introduced silhouette coefficients as an evaluation index that combines measures of intra-cluster cohesion and inter-cluster separation. This single metric provides a global assessment of clustering quality, primarily serving to assess the quality of clustering methods. The silhouette coefficient’s basic formula is as follows:(1)$$s(i)=\frac{b(i)-a(i)}{max(a(i),b(i))}$$

*a*(*i*) represents the average separation of sample *i* from other samples in the same cluster, which measures how closely the data point is related to others within its cluster.

*b*(*i*) denotes the average distance of sample *i* from samples in the nearest cluster that *i* is not a part of, assessing how well a point is separated from other clusters.

The silhouette coefficients of clustering are then determined by averaging these coefficients for each sample in the dataset. The resulting silhouette coefficient value falls within the range of 1 to −1, where better clustering quality is indicated by a larger value. Therefore, a higher silhouette coefficient indicates increased cluster cohesion and separation, reflecting a superior clustering outcome.

Table 9 displays silhouette coefficient values exceeding 0.8 for all three test images post clustering. Typically, a silhouette coefficient greater than 0.7 or 0.8 indicates a robust clustering effect. For image 1, the silhouette coefficient is 0.8783, signifying well-matched objects within clusters and significant separation from neighboring clusters. This high value surpasses the 0.8 threshold, reflecting strong cohesion within clusters and clear demarcation from others. Likewise, image 2 exhibits a silhouette coefficient of 0.8862, indicating well-defined clusters with tightly grouped objects. The substantial degree of separation from neighboring clusters suggests a reliable grouping of data points. Image 3 boasts a silhouette coefficient of 0.9157, representing an even higher level of clustering quality. Strong cohesion within clusters and remarkable separation from other clusters highlight the pronounced clustering effect, underscoring the algorithm’s robustness in delineating distinct groups within the data. In conclusion, the silhouette coefficients not only meet but exceed the commonly accepted threshold for robust clustering (0.8), emphasizing the efficacy of the algorithm. These findings provide a solid foundation for practical applications, particularly in concrete damage evaluation, where clustering quality directly influences the accuracy of insights derived from the data.

Table 9 also includes the percentage of detected cracks, allowing crack detection personnel to directly assess concrete damage severity based on crack prevalence. These data offer valuable insights for future evaluations, providing a comprehensive assessment of clustering quality and practical applications in concrete damage evaluation.

## 5. Conclusions

This study employed a thermal imaging camera to capture images of concrete cracks from two university buildings in South Korea, generating a database of 4500 mixed images to detect prominent concrete cracks in structural elements. The key findings and conclusions are summarized as follows:

ResNet50 neural network algorithm: A ResNet50 neural network algorithm for concrete crack detection was developed and implemented in Matlab R2020b. This model demonstrated both a fast detection speed and high accuracy. The application of edge algorithms such as Sobel and ROI techniques to subtract the background led to an enhanced output dataset containing crack information, achieving a remarkable accuracy of 99.9%.

Contour recognition with clustering: Clustering was implemented for the precise contour recognition of the identified images. Notably, the clustering algorithm was enhanced to self-optimize cluster values (k) and initial cluster centers. This optimization significantly improved the crack clustering accuracy.

Foundation for crack detection in real time: The algorithm proposed in this study serves as a foundational framework for advancing real-time crack detection, which is crucial for the continuous monitoring and maintenance of concrete structures. The proposed method empowers detection personnel to efficiently assess the extent of concrete damage.

In summary, the integration of a ResNet50 neural network, edge algorithms, and k-means clustering in this study presents a robust approach for accurate and efficient concrete crack detection. This method not only achieves high accuracy but also paves the way for real-world applications in concrete structure maintenance and real-time monitoring.

In the future, our emphasis will be on recognizing and continuously optimizing our model framework for the identification of crack width and depth. This includes assessing the model’s robustness under diverse conditions, such as evaluating its performance in various lighting conditions, different surface roughness levels, and various types of material damage. Additionally, we intend to investigate the impact of these factors on the technical specifications of the equipment.

## Figures and Tables

**Figure 1 sensors-24-01725-f001:**
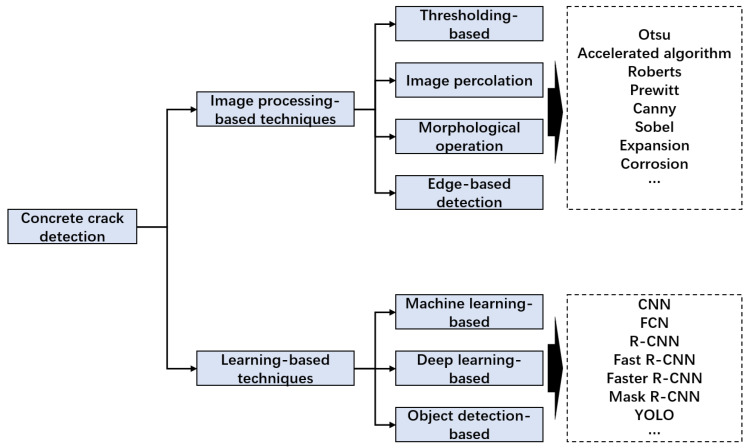
Classification of crack detection.

**Figure 2 sensors-24-01725-f002:**
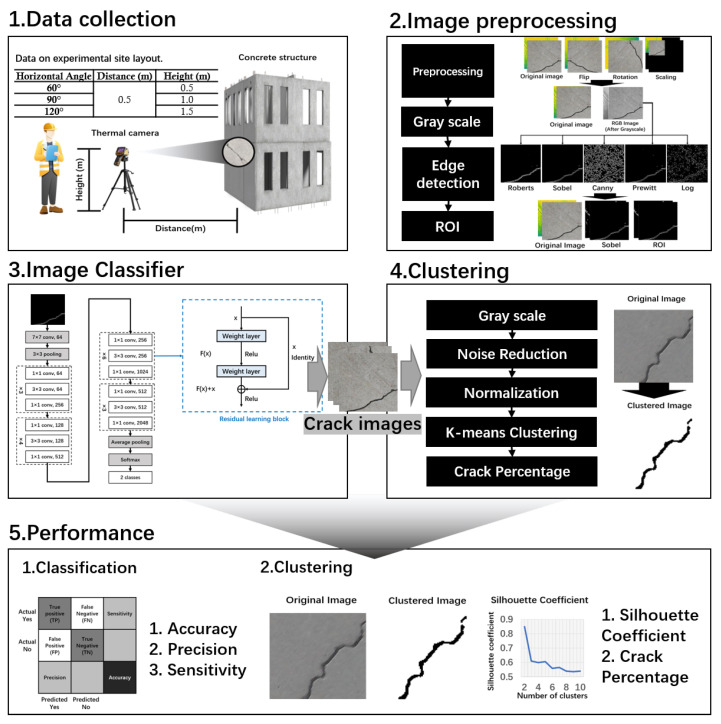
Research scheme.

**Figure 3 sensors-24-01725-f003:**
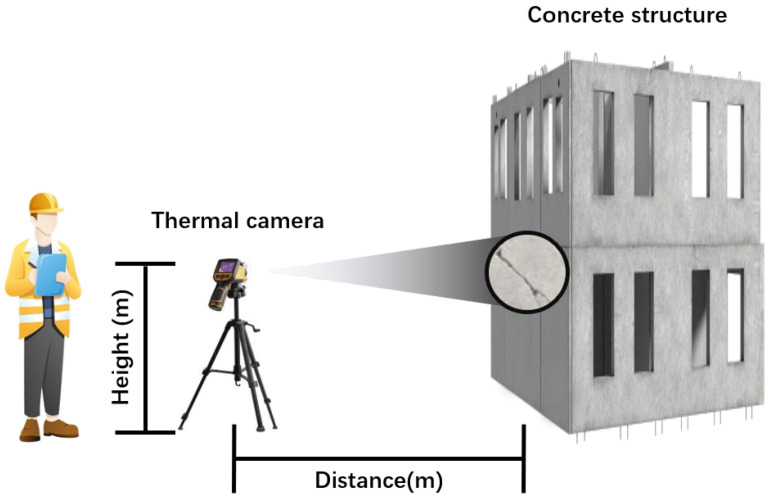
Experimental site layout.

**Figure 4 sensors-24-01725-f004:**
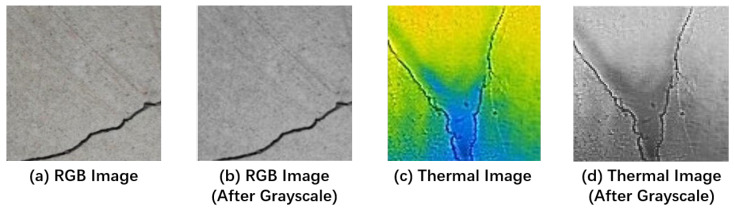
Samples of the dataset images: (**a**) RGB image, (**b**) RGB image (after grayscale), (**c**) thermal image, and (**d**) thermal image (after grayscale).

**Figure 5 sensors-24-01725-f005:**
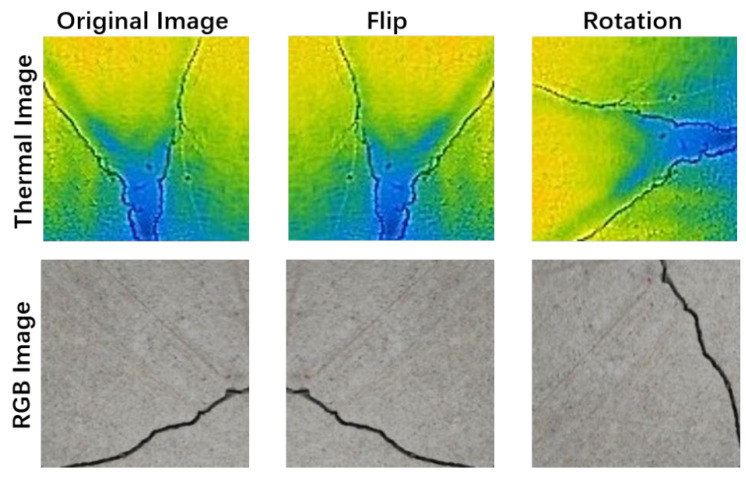
Preprocessing of the dataset images.

**Figure 6 sensors-24-01725-f006:**
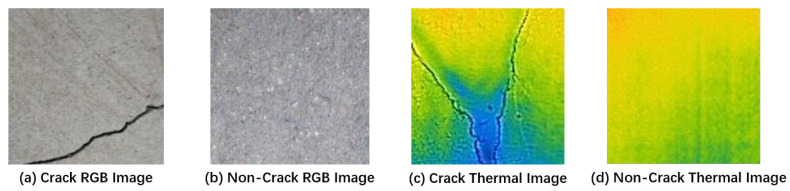
Grayscale of the dataset images: (**a**) crack RGB image, (**b**) non-crack RGB image, (**c**) crack thermal image, and (**d**) non-crack thermal image.

**Figure 7 sensors-24-01725-f007:**
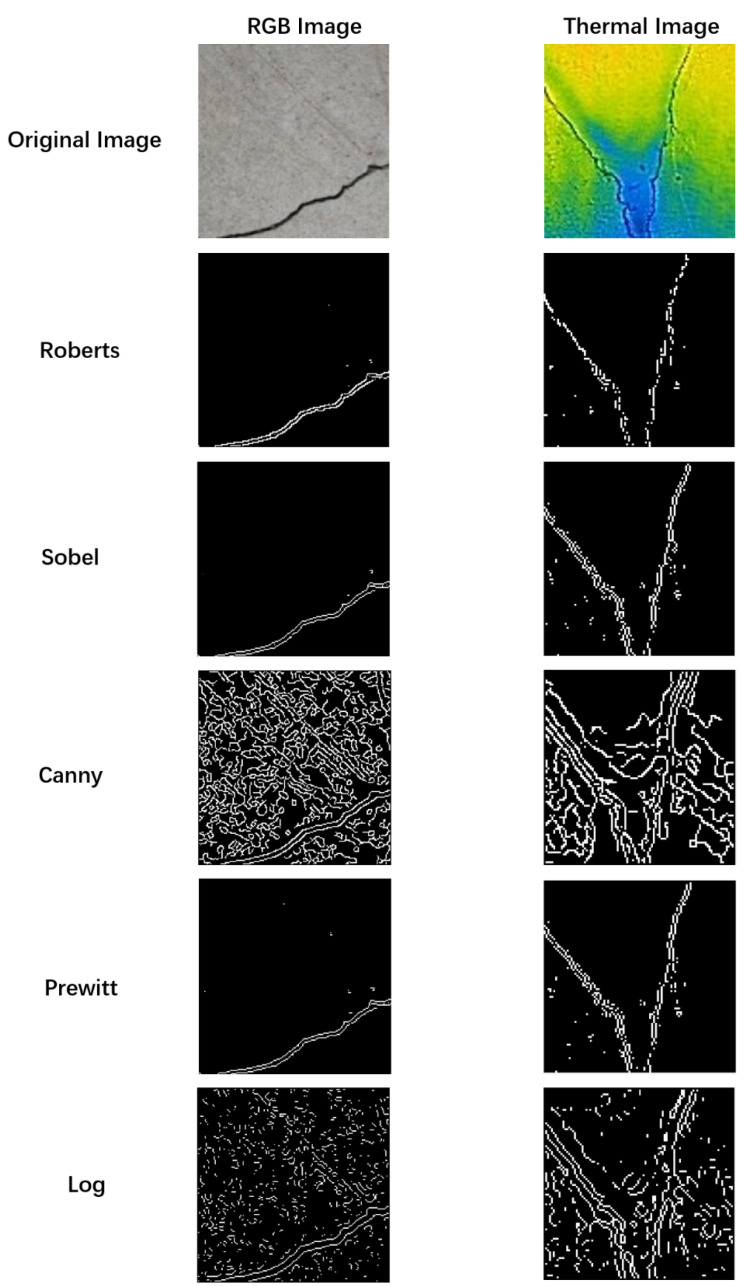
Edge detection of dataset images.

**Figure 8 sensors-24-01725-f008:**
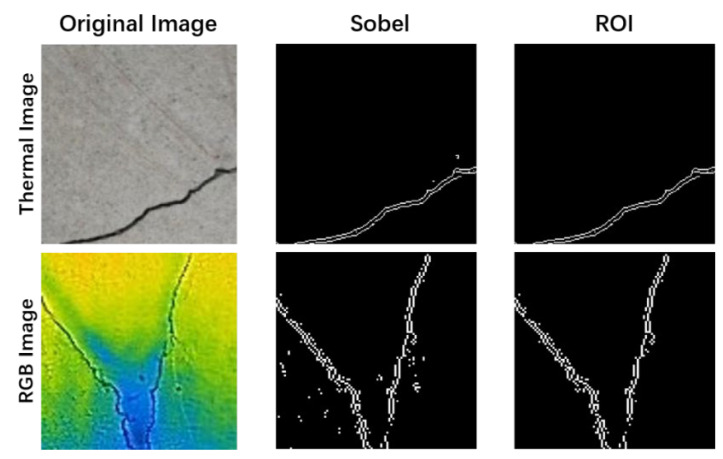
ROI of dataset images.

**Figure 9 sensors-24-01725-f009:**
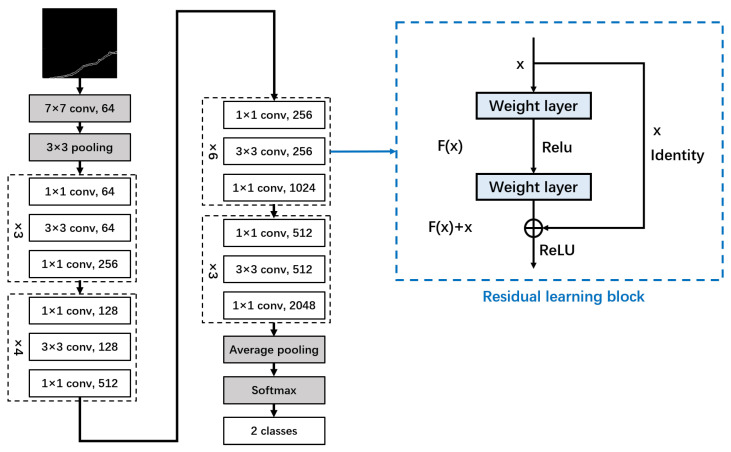
ResNet50 architecture (including residual learning block).

**Figure 10 sensors-24-01725-f010:**
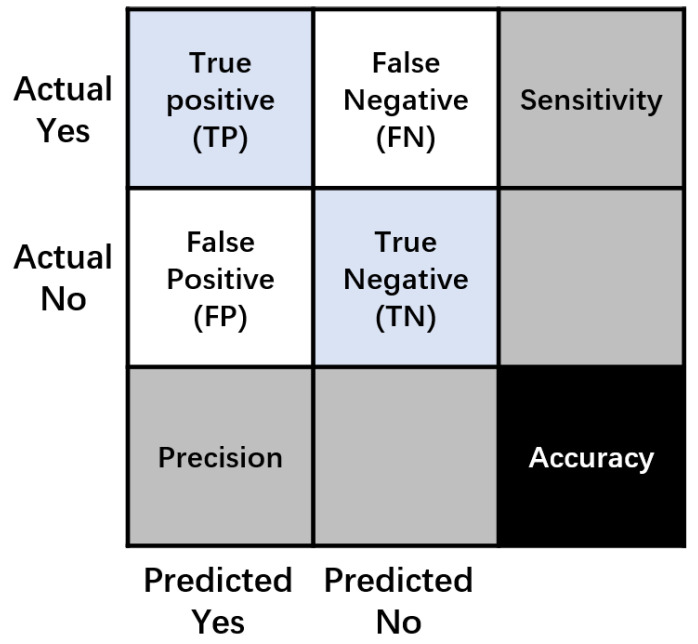
Confusion matrix.

**Figure 11 sensors-24-01725-f011:**
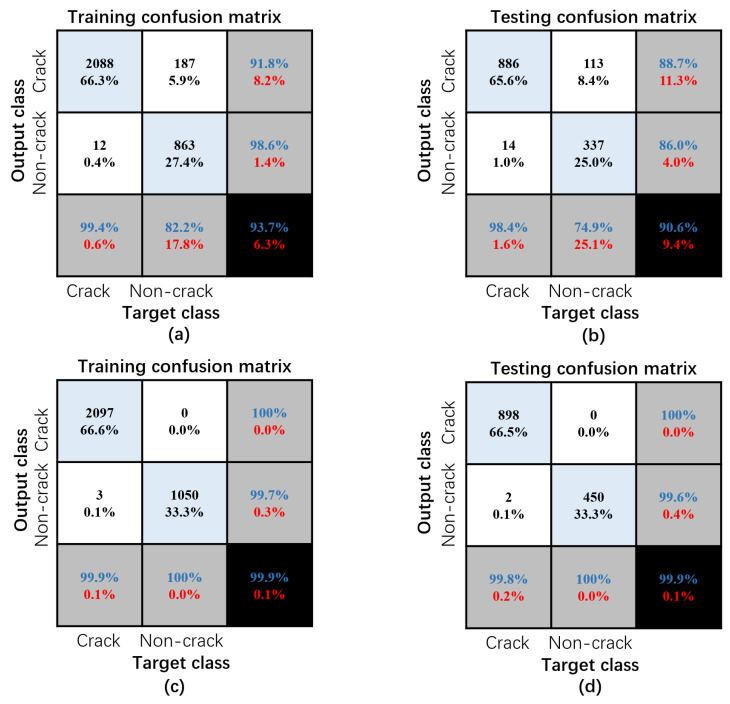
Training and testing confusion matrices: (**a**) Sobel training, (**b**) Sobel testing, (**c**) Sobel + ROI training, and (**d**) Sobel + ROI testing.

**Figure 12 sensors-24-01725-f012:**
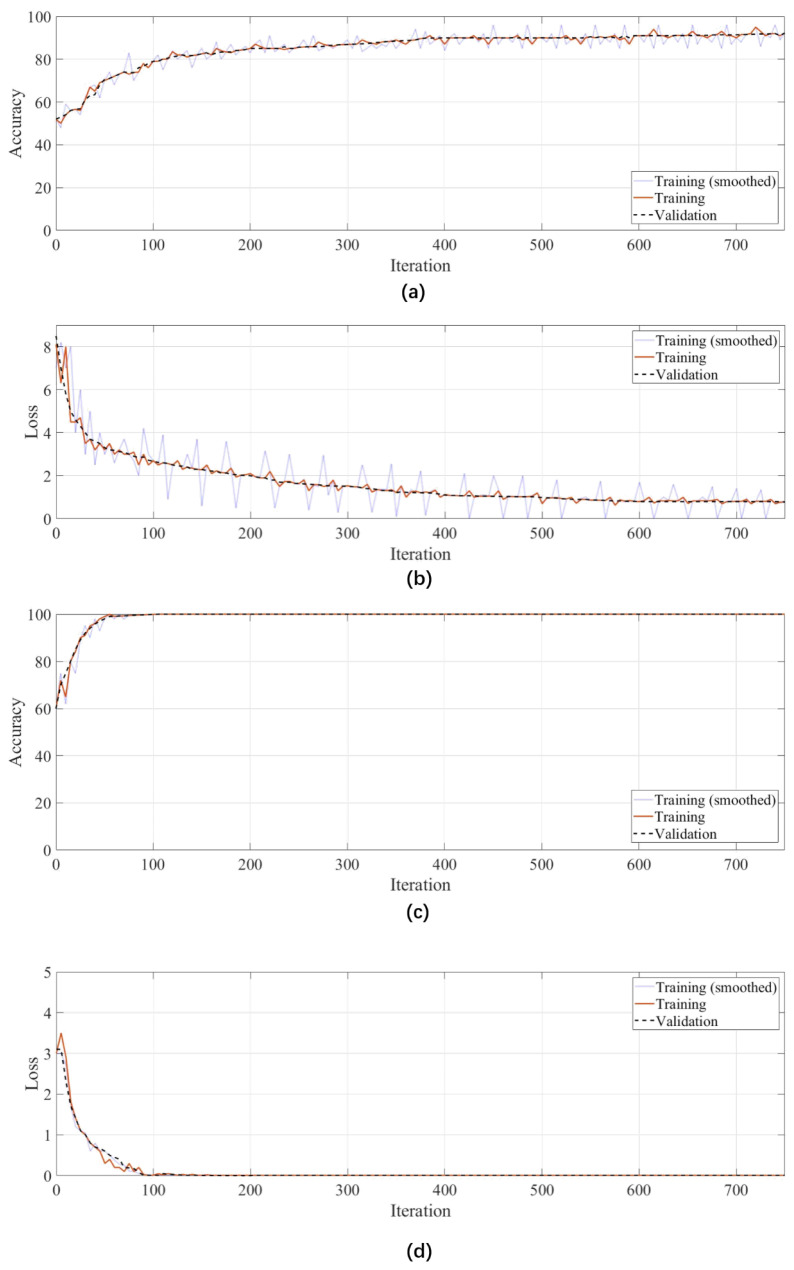
CNN training performance: (**a**) Sobel accuracy, (**b**) Sobel loss, (**c**) Sobel + ROI accuracy, and (**d**) Sobel + ROI loss.

**Figure 13 sensors-24-01725-f013:**
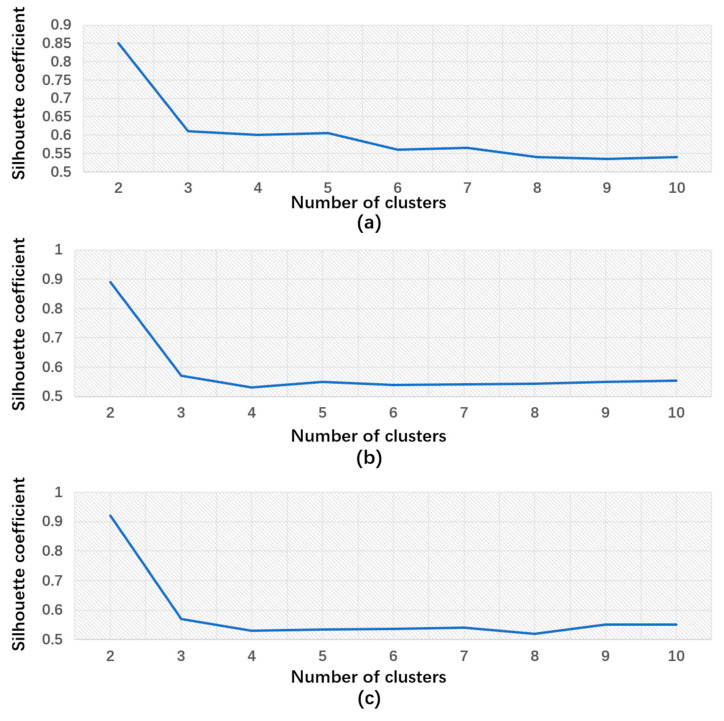
Silhouette scores for different cluster numbers: (**a**) image 1, (**b**) image 2, and (**c**) image 3.

**Figure 14 sensors-24-01725-f014:**
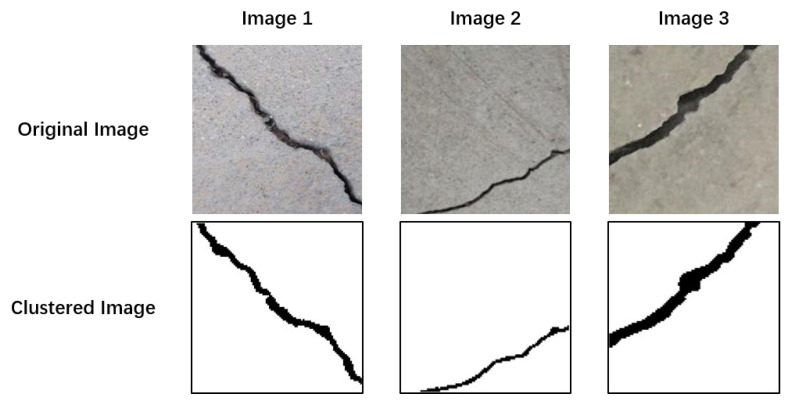
Clustering results.

**Table 1 sensors-24-01725-t001:** Technical parameters of thermal camera.

**Brand**	FLIR E8
**View Field**	45° × 34°
**Range of Temperatures**	−20 °C~250 °C
**Frequency Images**	9 Hz
**Temperature Sensitivity**	<0.06 °C
**Accuracy**	±2 °C
**Onboard Camera**	640 × 480

**Table 2 sensors-24-01725-t002:** Data on experimental site layout.

Horizontal Angle	Distance (m)	Height (m)
60°	0.5	0.5
90°		1.0
120°		1.5

**Table 3 sensors-24-01725-t003:** Dataset details.

	Total Images	Thermal Images	RGB Images
Non-Crack	1500	750	750
Crack	3000	1500	1500
Total	4500	2250	2250

**Table 4 sensors-24-01725-t004:** Confusion matrix for crack detection.

Total	Predicted Crack (PP)	Predicted Non-Crack (PN)
Actual crack (P)	TP	FN
Actual non-crack (N)	FP	TN

**Table 5 sensors-24-01725-t005:** Classifier model performance evaluation metric.

Metric	Formula
Accuracy (Acc)	$Accuracy=\frac{TP+TN}{TP+TN+FP+FN}$
Precision (Pr)	$Precision=\frac{TP}{TP+FP}$
Recall (Re)	$Recall=\frac{TP}{TP+FN}$
F1-Score (F1)	$F1-Score=\frac{2*Pr*Re}{Pr+Re}$

**Table 6 sensors-24-01725-t006:** Classifier model performance.

Concrete Crack and Non-Crack						
Neural Networks	Training			Testing		
	Precision	Recall	F1-Score	Precision	Recall	F1-Score
Sobel	99.4%	91.8%	95.4%	98.4%	88.7%	93.2%
Sobel + ROI	99.9%	100%	99.9%	99.8%	100%	99.8%

**Table 7 sensors-24-01725-t007:** Test clustered images.

Image 1	Image 2	Image 3
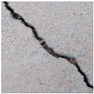	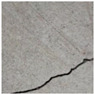	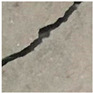

**Table 8 sensors-24-01725-t008:** Optimal cluster numbers and Otsu thresholds.

Crack Image	Optimal Number of Clusters	Otsu Threshold
Image 1	2	120
Image 2	2	116
Image 3	2	106

**Table 9 sensors-24-01725-t009:** Clustering model performance and crack percentage.

Crack Image	Silhouette Coefficient	Crack Percentage
Image 1	0.8783	6.13%
Image 2	0.8862	2.22%
Image 3	0.9157	7.32%

## Data Availability

The data and the code of this study are available from the corresponding author upon request.

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
