# Peer review of "Crack Detection and Analysis of Concrete Structures Based on Neural Network and Clustering"

_sensors, 2024, doi:10.3390/s24061725_

Round 1
Reviewer 1 Report
Comments and Suggestions for Authors
The manuscript presents a comprehensive study on employing a convolutional neural network (CNN) and clustering techniques for detecting cracks in concrete structures. The approach combines thermal imaging and ResNet50 architecture to achieve high accuracy in crack detection, enhanced by clustering methods for precise crack delineation. This study offers promising advancements in automated crack detection systems, crucial for infrastructure maintenance and safety. It fits well within the scope of Sensors, however, some aspects could be refined or expanded for greater impact:
- How does the model perform in real-world conditions where environmental variables (lighting, weather) might affect image quality?
- The manuscript would benefit from a more detailed comparison with current state-of-the-art methods. How does the proposed approach improve upon existing methods in terms of accuracy, efficiency, and computational resources?
- The study utilizes a dataset from two university buildings. Could the authors expand on the diversity of the dataset regarding different concrete textures, crack sizes, and types? This would help in understanding the model's applicability to a wider range of structures.
- Given the high accuracy of the model, can the authors provide insights into what features the CNN is focusing on for crack detection?
- The manuscript describes the use of k-means clustering for crack delineation. How sensitive is the model's performance to the choice of clustering algorithm? Would alternative clustering methods offer improvements or insights?
- The discussion on limitations and potential future directions could be expanded. Are there specific challenges the current model faces that could be addressed in future iterations?
- For practical application, what are the computational requirements for deploying this model in a real-time monitoring system?
- While the focus is on concrete structures, could the proposed method be adapted for other materials or structural elements?
- The manuscript presents qualitative results from the clustering process. Could the authors include quantitative metrics (e.g., silhouette score) to evaluate the clustering performance systematically?
Author Response
Dear reviewers.
Relevant reply documents and content-updated manuscripts have been uploaded simultaneously.
Thank you.

Reviewer 2 Report
Comments and Suggestions for Authors
1. The structure of the Introduction section of the paper is disorganized. It is recommended to merge Chapters 1 and 2 to provide a focused overview of the research aim, questions, the current state of research, and the content of this paper’s study.
2. In the Introduction section, the authors provide an inadequate presentation of the current developments in structural health monitoring (SHM) for that year; moreover, there is a lack of introduction to the hyperparameter optimization of the CNN network structure. It is recommended that the author address this. Some topics should be considered in the section of Introduction, such as the development of SHM (The Current Development of Structural Health Monitoring for Bridges: A Review), and hyperparameter (Nonlinear modeling of temperature-induced bearing displacement of long-span single-pier rigid frame bridge based on DCNN-LSTM).
3. In Figure 6, the four images should have sub-caption numbers such as (a), (b), (c), and (d). The authors are advised to revise this and check the sub-captions for the remaining figures.
4. The paper contains many explanatory sentences, for example, from Line 306 to 311. It is not necessary to explain the concepts of gradient disappearance and gradient explosion, as the authors can easily find this information through a Google search if needed. It is suggested that the authors simplify the articulation of the paper.
5. In Figure 9, it is advised that ‘Relu’ be updated to ‘ReLU’.
6. The hyperparameters of the CNN structure are crucial; it is recommended that the authors provide a detailed introduction to the specific settings of the hyperparameters, and discuss the potential for optimization, in order to enhance the reader’s understanding of the paper’s key techniques.
7. In Figure 13, it is suggested to include a vertical axis.
8. The paper uses thermal imaging cameras to obtain images of the structure’s surface. Can factors such as ambient temperature and noise (Two-stage damage identification for bridge bearings based on sailfish optimization and element relative modal strain energy) affect the accuracy of identification? The authors are advised to address this in the paper.
9. The paper presents a method for crack identification, which can determine the presence of cracks on the surface of buildings. In damage identification, the existence of cracks is the first stage, the location of the cracks is the second stage, and the width and depth of the cracks represent the third stage. Can the method presented in the paper ascertain the exact location and width of the cracks, or even their depth?
Author Response

(The authors gave the same response as above.)

Round 2
Reviewer 1 Report
Comments and Suggestions for Authors
All comments are implemented.
Author Response
Thank you very much for your feedback and help.
Reviewer 2 Report
Comments and Suggestions for Authors
1. There is a lack of introduction to the hyperparameter optimization of the CNN network structure. It is recommended that the author address this.
2. The paper used thermal imaging cameras to obtain images of the structure’s surface. Can factors such as ambient temperature and noise that affect the accuracy of identification? The authors are advised to address this in the paper.
Author Response
Dear reviewers, the pdfs of the relevant questions and answers have been synchronized and uploaded.
Thank you for your feedback!
